# Anti-Periodontitis Effect of Ethanol Extracts of *Alpinia Katsumadai* Seeds

**DOI:** 10.3390/nu14010136

**Published:** 2021-12-28

**Authors:** Seo Woo Shin, Young Sun Hwang

**Affiliations:** Department of Dental Hygiene, College of Health Science, Eulji University, 553, Sansung-Daero, Soojung-Gu, Seongnam City 13135, Korea; kiteys@eulji.ac.kr

**Keywords:** ethanol extract of *Alpinia Katsumadai* seeds, periodontitis, dental plaque bacteria, lipopolysaccharide, inflammation, bone resorption

## Abstract

Oral microbes are intimately associated with many oral and systemic diseases. Ongoing research is seeking to elucidate drugs that prevent and treat microbial diseases. Various functions of *Alpinia Katsumadai* seed extracts have been reported such as their anti-viral, anti-oxidant, anti-inflammatory, anti-puritic, anti-emetic, and cytoprotective effects. Here, we investigated the anti-periodontitis effect of an ethanol extract of *Alpinia Katsumadai* seeds (EEAKSs) on dental plaque bacteria (DPB)-induced inflammation and bone resorption. DPB and *Porphyromonas gingivalis* (*P. gingivalis*) were cultured and lipopolysaccharide (LPS) was extracted. Prostaglandin E_2_ (PGE_2_) and cyclooxygenase 2 (COX-2) levels were estimated using ELISA. Cytotoxicity was also verified. Proteases were screened using a protease antibody array method. Osteoclastic bone resorption was also investigated. EEAKSs suppressed *P. gingivalis* growth on agar plates. LPS prepared from dental plaque bacteria (DPB-LPS) and *P. gingivalis* (PG-LPS) significantly increased PGE_2_ and COX2 levels in immortalized gingival fibroblasts (IGFs), immortalized human oral keratinocytes (IHOKs), and RAW264.7 macrophage cells. However, DPB-LPS and PG-LPS-induced PGE_2_ and COX-2 increases were effectively abolished by EEAKS treatment at non-cytotoxic concentrations. In the protease antibody array, matrix metalloproteinase (MMP)-2, MMP-3, MMP-7, kallikrein 10, cathepsin D, and cathepsin V levels were increased by PG-LPS stimulation. However, increases in protease levels except for cathepsin D were suppressed by EEAKS treatment. In addition, RANKL-induced osteoclast differentiation was significantly inhibited by EEAKS treatment, leading to reductions in resorption pit formation. These results suggest that EEAKSs exerted a beneficial oral health effect to help prevent DPB-mediated periodontal disease.

## 1. Introduction

Periodontal disease is a highly prevalent disease that occurs in over 90% of adults and is the main cause of tooth loss [1]. It is divided into gingivitis and periodontitis according to the severity of the disease. Gingivitis is a periodontal disease that can be quickly resolved and is limited to gum inflammation. However, periodontitis is an inflammatory lesion of periodontal tissue caused by proteolytic dental plaque bacteria present in the gingival margin. Therefore, a periodontal pocket is formed, gingival retraction occurs, and periodontal ligaments and alveolar bone are destroyed, causing tooth loss. Mixed infections with various bacteria present in dental plaque cause the inflammation of the periodontal tissue. In particular, *Porphyromonas gingivalis* (*P*. *gingivalis*), which is present in oral biofilms and belongs to the red complex, has one of the highest risks for causing periodontal disease [2]. As inflammation progresses, *P. gingivalis* infiltrates into the gingival sulcular epithelium and is easily detected in the lesions of patients with periodontitis. Periodontal tissue is disrupted by metabolites, toxins, and proteolytic enzymes secreted by *P. gingivalis* in the process of attaching and penetrating epithelial cells and is also destroyed by cytokines produced by the host cells that react with *P. gingivalis*.

Lipopolysaccharide, a structural component of the cell wall of Gram-negative bacteria, directly destroys tissues by acting as an endotoxin and stimulates the immune system to produce free radicals, prostaglandins, and various cytokines, causing inflammation [3]. These inflammatory mediators stimulate the secretion of collagenase from periodontopathogenic bacteria to decompose collagen in the periodontal tissues matrix, leading to the regression of the gums [4]. Therefore, suppressing the production of inflammatory mediators and proteolytic enzymes is very important for preventing periodontal disease. Recently, research on natural products for the prevention and treatment of periodontal disease has steadily progressed. *Alpinia Katsumadai* seeds are plants belonging to the ginger family. The peeled seeds are widely used as an anti-emetic medicine and to treat gastric disorders in oriental medicine [5]. The anti-oxidant, anti-viral, anti-asthmatic, and cytoprotective effects of *Alpinia Katsumadai* seeds have also been reported [5,6]. Ingredients such as diaryiheptanoids, flavonoids, monoterpenes, sesquiterpenoids, and stilbenes are known as the components of *Alpinia Katsumadai* seeds [7]. However, little is known about the intracellular events involved in the therapeutic effect of *Alpinia Katsumadai* seeds. In particular, the effect of *Alpinia Katsumadai* seeds on the inflammatory response is unknown.

In this study, we prepared an ethanol extract of *Alpinia Katsumadai* seeds (EEAKSs) and studied the effect of EEAKSs on periodontitis. For this, we verified the anti-bacterial effect of EEAKSs against dental plaque bacteria and *P. gingivalis*. The protective effect of EEAKSs on inflammation and osteoclastic bone resorption induced by dental plaque bacteria–fLPS (DPB-LPS) or *P. gingivalis*–LPS (PG-LPS) was also investigated. This result suggests that EEAKSs can be an effective agent in the prevention and treatment of periodontitis.

## 2. Materials and Methods

### 2.1. Cell Lines and Culture Media

Immortalized human gingival fibroblasts (IGFs) and immortalized human oral keratinocytes (IHOKs) were kindly provided by the Yonsei University College of Dentistry, Oral Cancer Institute (Seoul, Korea) and cultured in Dulbecco’s modified Eagle’s medium (DMEM)/F-12 medium (3:1 ratio) with 10% fetal bovine serum as previous detailed [8]. RAW264.7 macrophage cells were cultured in RPMI 1640 containing 10% fetal bovine serum (Gibco BRL). 

### 2.2. Bacterial Culture

The *Pophyromonas gingivalis* strain ATCC 33277 was purchased from ATCC (Manassas, VA, USA) and cultured in Wilkins–Chalgren anaerobe broth (KisanBio, Seoul, Korea) and on Wilkins–Chalgren agar (KisanBio, Seoul, Korea) at 37 °C in an anaerobic incubator (80% N_2_, 10% CO_2_, and 10% H_2_ gas mix). Aliquots of 50 µL bacterial culture were inoculated into 5 mL of medium under anaerobic conditions and incubated overnight before experiments. An anaerobic chamber (Modular incubator chamber, MIC-101, Billups-Rothenberg Inc., Shirley, NY, USA) was used to create anaerobic environment. An optical density of the bacterial culture was approximately 0.8 (600 nm). For agar plate cultures, a bacterial suspension was inoculated onto the top agar and bacterial growth was assessed. Dental plaque bacteria from dental plaques were cultured in brain–heart infusion (BHI) broth (Becton, Dickinson and Company, Baltimore, MD, USA) at 37 °C. Dental plaques were obtained by a dental hygienist through regular scaling from a participant. Informed consent was given by the participant.

### 2.3. Ethanol Extracts of Alpinia Katsumadai Seeds

The ethanol extracts of *Alpinia Katsumadai* seeds (EEAKSs) was provided by COSMAX Inc. R&I Center (Seongnam, Korea). *Alpinia Katsumadai* seeds were ground to a fine powder and extracted material in 70% ethanol. A voucher specimen was deposited at the by COSMAX Inc. R&I Center (Seongnam, Korea). 

### 2.4. Lipopolysaccharide Extractions from Dental Plaque Bacteria

An LPS extraction kit (iNtRON Biotechnology, Seongnam, Korea) was used for lipopolysaccharides (LPSs) extraction from dental plaque bacteria according to the manufacturer’s manuals. The Limulus Amebocyte Lysate Chromogenic Endotoxin Quantitation Kit (Pierce Biotechnology, Rockford, IL, USA) was used for LPS quantification. *Escherichia coli* O111:B4 was used as the concentration standard. Ten endotoxin units (EU)/mL equaled approximately 1 ng/mL.

### 2.5. Enzyme-Linked Immunosorbent Assay

The level of human Prostaglandin E_2_ (PGE_2_) and human cyclooxygenase-2 (COX-2) was estimated by an ELISA kit. The following ELISA kits were purchased from their respective sources: human PGE_2_ (KGE004B; R&D Systems, Minneapolis, MN, USA), human COX-2 (ab267646; Abcam). Moreover, 1 × 10^5^ cells were cultured in media with DPB-LPS and/or EEAKSs for 24 h. The medium was used for analysis.

### 2.6. Cytotoxicity Assay

Cytotoxicity was tested with 3-(4,5dimethylthiazol-2-yl)-2,5-diphenyltetrazolium bromide (MTT) assay (Sigma-Aldrich, St Louis, MO, USA). Briefly, 1 × 10^4^ cells was cultured with complete medium with or without EEAKSs for 24 h. Cell medium was replaced with medium containing 5 mg/mL MTT and further cultured for 2 h. Moreover, 100 µL dimethyl sulfoxide (DMSO) was added to dissolve the purple product. The absorbance was measured at 570 nm using a microplate reader (SynergyTM HTX Multi-Mode Microplate Reader; BioTek Instruments Inc., Winooski, VT, USA).

### 2.7. Protease Antibody Array 

The Proteome Profiler Human Protease Array Kit (R&D Systems Inc., Minneapolis, MN, USA) was used for the antibody array. Among the confluence cells, 70% were cultured in serum-free medium containing DPB-LPS and/or EEAKSs for 24 h. The culture medium was harvested by centrifugation at 1000 rpm for 5 min and was used for conditioned medium (CM). Protease antibody array membrane was incubated with CM overnight and the following procedure was performed according to the manufacturer’s protocols. Relative enhanced chemiluminescence (ECL) signal levels pf proteases were compared using Image J program (National Institutes of Health, Bethesda, MA, USA).

### 2.8. Osteoclast Formation

Mouse bone marrow-derived macrophages (BMMs) were prepared from the tibias of 4-week-old ICR mice using Histopaque and cultured in minimum essential medium alpha medium (α-MEM) containing 10% FBS and M-CSF (30 ng/mL; R&D System, Minneapolis, MN, USA). Moreover, 5 × 10^4^ mouse BMMs cells were cultured in M-CSF (30 ng/mL), recombinant mouse soluble RANK ligand (sRANKL) (100 ng/mL; Koma Biotech, Seoul, Korea), DPB-LPS (1 µg/mL), and/or EEAKSs (10 µg/mL) for 10 days with replacement with fresh medium. After fixation, cells were reacted with the acid phosphatase, leukocyte (TRAP) kit (Sigma-Aldrich, St. Louis, MO, USA) according to the manufacturer’s manuals. TRAP-positive multinucleated cells (≥3 nuclei) were imaged and counted.

### 2.9. Pit Formation 

Osteo assay surface polystyrene stripwells (Corning, Corning, NY, USA) were used to observe the pit formation by functional osteoclast. The 5 × 10^4^ mouse BMMs cells prepared from the tibias of ICR mice were cultured in α-MEM media containing M-CSF (30 ng/mL), sRANKL (100 ng/mL), DPB-LPS (1 µg/mL), and/or EEATSs (10 µg/mL) for 2 weeks. Culture medium was replaced with fresh medium. Sodium hypochlorite solution was then added to the remaining lysis cells. The resorption pit was washed with PBS and images were captured. 

### 2.10. Statistical Analysis

InStat GraphPad Prism ver. 5.01 statistical software (GraphPad Software, Inc., San Diego, CA, USA) was used for statistical analyses. Non-parametric Kruskal–Wallis tests with Dunn’s post hoc analysis was employed for multiple comparisons. The data are expressed as the mean ± standard error of the mean (SEM). *p* < 0.05 was considered statistically significant.

## 3. Results

### 3.1. Effects of EEAKSs on P. gingivalis Growth

*Porphyromonas gingivalis* is a major pathogenic bacterium of chronic periodontitis [9]. To test the effect of EEAKSs on *P. gingivalis* growth, cells were cultured until the optical density was approximately 0.8 (600 nm) and 10 µL of bacterial solution was inoculated onto an agar plate, and cultured with various concentrations of EEAKSs for 48 h. As shown in Figure 1, EEAKSs effectively suppressed *P. gingivalis* growth. As the concentration of the EEAKSs treatment increased, the number of colonies significantly decreased. 

### 3.2. Effect of EEAKSs on Bacterial LPS-Induced PGE_2_ and COX-2 Levels

The synthesis of PGE_2_ begins with the production of arachidonic acid from membrane phospholipids by the enzymatic reaction of phospholipase A_2_, which is catalyzed by cyclooxygenase (COX). COX-2 is involved in both acute and chronic inflammatory response by producing PGE_2_ [10]. An increased PGE_2_ level was observed in gingival crevicular fluids of patients with periodontal disease [11]. To observe the effect of EEAKSs on inflammation, we first prepared bacteria (DPB) from dental plaques through regular scaling (*n* = 3), and the LPS of DBP (DPB-LPS) was extracted. LPS from *P*. *gingivalis* (PG-LPS) was also prepared by the same extraction protocol. Then, we tested the effect of DPB-LPS and PG-LPS on the PGE_2_ levels of various types of human cell lines. Moreover, 1 µg/mL of each DPB-LPS-1, DPB-LPS-2, DPB-LPS-3, or PG-LPS was treated to media for 24 h and ELISA analysis was performed. As shown in Figure 2A, a significant increase in PGE_2_ levels by DPB-LPS was observed in the IGF, IHOK, and RAW264.7 cells. COX-2 levels were also increased by stimulation with 1 µg/mL DPB-LPS-1, DPB-LPS-2, DPB-LPS-3, or PG-LPS. The increases in PGE_2_ and COX-2 levels by DPB-LPSs or PG-LPS were different in function of the cell line. DPB-LPS-2 induced the most effective inflammatory response and was used for further studies (DPB-LPS). Then, the cells were treated with EEAKSs and the levels of PGE_2_ and COX-2 were analyzed. As shown in Figure 2B, the PGE_2_ and COX-2 levels increased by DPB-LPS or PG-LPS were significantly decreased by the 10 µg/mL EEAKS treatment in the various types of cells. To investigate the effect of EEAKSs on the viability of IGF, IHOK, and RAW264.7 cells, the MTT assay was performed. No apparent cytotoxicity in IGF, IHOK, and RAW264.7 cells was observed at ≤20 µg/mL EEAKSs, but weak cell growth inhibition was observed at ≥25 µg/mL EEAKSs (Figure 2C). Thus, EEAKSs not only inhibited the growth of dental plaque bacteria and *P*. *gingivalis* through its anti-bacterial effect but also inflammation by suppressing bacterial-induced PGE_2_ and COX-2 expression, thereby providing a beneficial effect in the prevention and management of periodontitis. 

### 3.3. Identify the Proteases Involved in PG-LPS-Induced Inflammation

Inflammation is a defense mechanism to protect tissues from infection or injury, and proteolytic activity is induced to remove and recover damaged tissues. However, proteases secreted by bacteria in the periodontal pocket cause periodontitis by destroying the surrounding tissues and causing alveolar bone resorption [12]. To elucidate the proteases involved in *P*. *gingivalis*-induced inflammation, IHOK cells were treated with 1 µg/mL PG-LPS with or without 10 µg/mL EEAKSs and the protease antibody array assay was performed using the culture medium. In only the PG-LPS-treated IHOK cells, the proteases profiles were increased for matrix metalloproteinase (MMP)-7 (➀), Kallikrein 10 (②), Cathepsin V (③), Cathepsin D (④), MMP-2 (⑤), and MMP-3 (⑥) by 1.6~2.5-fold compared to the dimethyl sulfoxide (DMSO) control (Figure 3). However, the increased protease levels were significantly decreased by the EEAKS treatment—except for cathepsin D. 

### 3.4. Effects of EEAKSs on RANKL-Induced Osteoclast Formation and Bone Resorption

Alveolar bone loss is a hallmark of periodontitis progression [13]. To observe the effect of EEAKSs on osteoclastic bone resorption, bone marrow-derived macrophage (BMM) cells were treated with RANKL with or without EEAKSs and osteoclast differentiation was observed. TRAP-positive multinucleated osteoclasts were clearly detected in the medium containing 100 ng/mL of RANKL (Figure 4). However, EEAKSs significantly inhibited RANKL-induced osteoclast formation in a dose-dependent manner. Significant reductions in osteoclast numbers were observed by treatment with ≥5 mg/mL EEAKSs. The bone resorption activity of mature osteoclasts was also measured by the pit formation assay. Consistent with the osteoclast formation studies, resorption pits induced by RANKL were significantly inhibited by EEAKS treatment. These results indicate that EEAKSs inhibited RANKL-induced osteoclast formation and bone resorption.

## 4. Discussion

Periodontitis is a highly prevalent periodontal disease found in 20–50% of the global population [14]. In the Republic of Korea, gingivitis and periodontal disease have remained the second most frequent diseases over the past decade. In particular, as the prevalence of periodontal disease increases with age, periodontal disease is becoming an obstacle in nutritional management to maintain individual health and prevent diseases. Tooth loss is strongly associated with malnutrition in older adults [15]. Therefore, research has been conducted to derive effective ingredients for the prevention and treatment of periodontal disease, and many natural ingredients have been suggested. Herbal extracts, catechin, theaflavin, curcumin, and garlic components were reported to be effective ingredients for periodontitis. In gingival cells, herbal extract, theaflavin, and curcumin inhibited PGE_2_, IL-6, IL-8, and tumor necrosis factor (TNF)-a levels [16,17,18,19,20]. In animal studies, sumac extract, epicatechin gallate, epigallocatechin gallate, and curcumin derivative inhibited the alveolar bone resorption [21,22,23,24]. Garlic ingredients such as allicin, diallyl sulfide, and aged garlic extract also inhibited the growth of periodontitis-causing bacteria such as *P. gingivalis*, *A. actinomycetemcomitans*, and *F. nucleatum* [25,26,27,28].

Although *Alpinia katsumadai* seeds have various effects such as anti-oxidant and anti-viral, their role in inflammatory response by endogenous oral bacteria has not been reported. In this study, we sought to confirm the effect of *Alpinia katsumadai* seed extracts on the inflammatory response caused by periodontopathic bacteria. In addition, we also tried to confirm the effect of *Alpinia katsumadai* seed extracts on alveolar bone resorption observed in aggressive periodontitis. To test its anti-bacterial activity, *P. gingivalis* was cultured on agar plates containing different concentrations of EEAKSs. *P. gingivalis* growth was significantly inhibited by EEAKSs in a dose-dependent manner. This result encouraged us to study the role of EEAKSs in the inflammatory response and alveolar bone resorption observed in periodontitis. For these studies, dental plaque bacteria (DPB) were cultured from dental plaques through regular scaling, and whether DPB induced inflammation in various human cell lines was analyzed. However, in vitro co-cultures of human cell lines and DPB caused pH changes in culture medium during overnight incubation. LPS was therefore extracted from DPBs (DPB-LPS), and its capacity to induce inflammation was verified. LPS from *P. gingivalis* (PG-LPS) was also prepared by the same extraction procedure. LPS extracted from bacteria was quantified using the Endotoxin Quantitation Kit for further studies. In immortalized gingival fibroblasts (IGFs), immortalized human oral keratinocytes (IHOKs), and RAW 264.7 macrophages, 1 µg/mL of DPB-LPS and PG-LPS significantly increased the PGE_2_ and COX-2 levels. There was a difference in the stimulation degree according to the three extracted LPSs, but strong PGE_2_ and COX-2 stimulation by DPB-LPS or PG-LPS was observed compared to the controls. However, the PGE_2_ and COX-2 secretion increased by DPB-LPS or PG-LPS was significantly decreased by EEAKS treatment and the inhibitory effect was dose-dependent. The concentrations of EEAKSs (5–10 µg/mL) that inhibited bacterial growth and PGE_2_ and COX-2 secretion did not induce cytotoxicity in IGFs, IHOKs, and RAW 264.7 cells. These results suggest that EEAKSs are a useful agent for inhibiting periodontitis-causing bacterial growth without cytotoxicity.

In periodontitis, many types of protease secretion are increased and play roles in matrix destruction and alveolar bone regression [29]. To identify the proteases induced by PG-LPS, culture media from IHOK cells treated with PG-LPS and/or EEAKSs for 24 h were analyzed using the human protease antibody array kit in duplicate with 34 different protease antibodies. Among the proteases increased by PG-LPS, MMP-2, MMP-3, MMP-7, kallikrein 10, and cathepsin V levels were decreased by EEAKS treatment. MMPs are proteolytic enzymes involved in the degradation of the extracellular matrix of various tissues including bones [30]. Kallikrein is involved in bone resorption by the kallikrein–kinin system and the coagulation cascade [31]. Cathepsins play roles in lysosomal protein turnover, which contributes to a plethora of physiological processes, such as antigen presentation, bone remodeling, and epidermal homeostasis [32]. The induction of MMP-2 and MMP-3 in osteoblasts is essential for bone resorption [33]. Additionally, MMP-7 promotes prostate cancer-induced osteolysis via the solubilization of the receptor activator of nuclear factor-kappaB (RANKL) [34]. Although the protein level of kallikrein 10 may change during disease initiation and progression, kallikrein 10 overexpression is correlated with breast cancer aggressiveness [35]. Cathepsin V has potent elastolytic activity and accelerates the destruction of the elastin matrix in diseased arteries [36]. Osteoclast differentiation and bone resorption activity require stimulation by RANKL expressed on osteoblasts [37]. In this study, we also observed the role of EEAKSs in osteoclastogenesis. In the RANKL-induced osteoclast differentiation process, EEAKS treatment significantly inhibited osteoclast formation in a dose-dependent manner. Taken together, the bone resorption activity of mature osteoclasts was also suppressed by EEAKS treatment. These results suggest that EEAKSs could be an effective ingredient in controlling proteases secretion and alveolar bone regression caused by periodontitis. In this study, the antibacterial and anti-inflammatory activity of EEAKSs against anaerobes was confirmed using the anaerobic *P. gingivalis* (Figure 5). However, in the case of dental plaque bacteria, these activities were only observed for bacteria cultured under aerobic conditions. It is a limitation of our study that antibacterial and anti-inflammatory activities were not observed against anaerobic bacteria in dental plaque. In addition, if analytical grade EEAKSs are available, a further study to compare the activity using them as an experimental control will be needed.

## 5. Conclusions

Periodontitis is an aggressive bacterial inflammatory disease and *P*. *gingivalis* is the representative causative bacterium. EEAKSs effectively inhibit *P. gingivalis* growth and abolish bacterial LPS-induced proteases levels. In addition, EEAKSs are an effective plant ingredient that can regulate the expression of inflammatory factors caused by bacterial LPS. Therefore, EEAKSs can be suggested as an antibacterial and anti-inflammatory agent for preventing and controlling periodontitis. Further studies are needed to isolate the functional components of the extract and to determine its delicate activity for pharmaceutical use.

## Figures and Tables

**Figure 1 nutrients-14-00136-f001:**
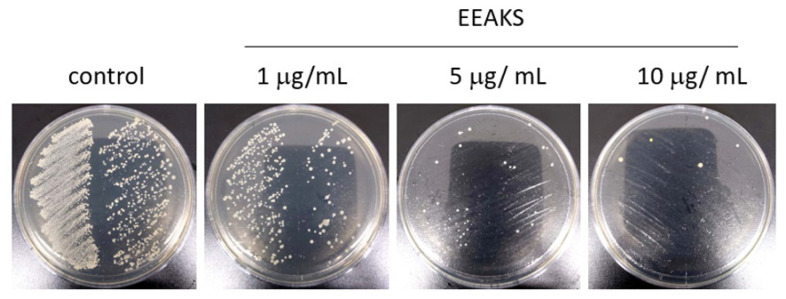
EEAKSs suppress *Porphyromonas gingivalis* growth. The effect of EEAKSs on *P*. *gingivalis* growth was investigated on Wilkins–Chalgren agar plates. *P. gingivalis* were cultured until optical density was approximately 0.8 (600 nm) and 10 µL of bacterial solution was cultured on an agar plate with or without EEAKSs for 48 h in anaerobic chamber. Representative images are shown.

**Figure 2 nutrients-14-00136-f002:**
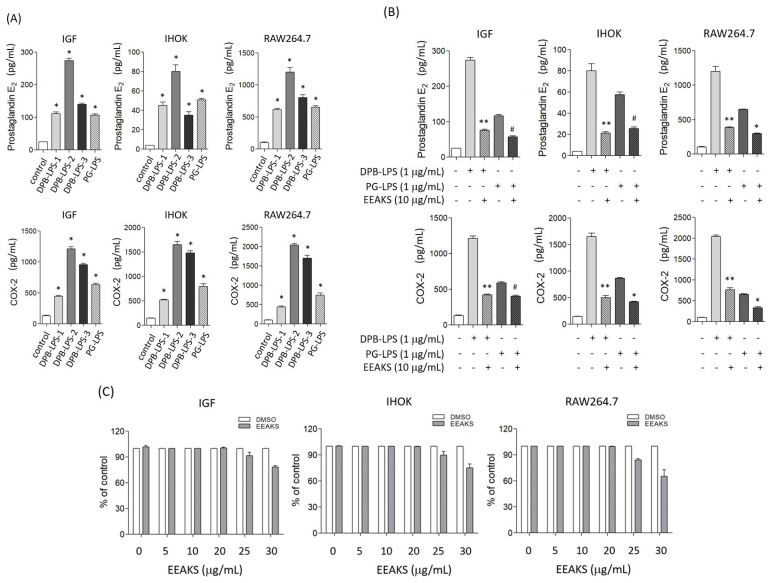
Effect of the EEAKSs on dental plaque bacterial-LPS-induced prostaglandin E_2_ and cyclooxygenase-2. (**A**) The effect of bacterial LPS on PGE_2_ and COX-2 levels in immortalized gingival fibroblast (IGF), immortalized human oral keratinocyte (IHOK), and RAW264.7 macrophage. Three DPB-LPS (DPB-LPS-1, DPB-LPS-2, DPB-LPS-3) were prepared from dental plaques and each DPB-LPS (1 µg/mL) was treated to media for 24 h and PGE_2_ and COX-ELISA analysis was performed. PG-LPS was also extracted from *P*. *gingivalis*. Moreover, 1 µg/mL PG-LPS was treated to media for 24 h and ELISA analysis was also performed. The data are expressed as the mean ± standard error of the mean (SEM). ^#^
*p* < 0.01, * *p* < 0.001 vs. without bacterial LPS control medium from each cell line. (**B**) The effect of EEAKSs on bacterial LPS-induced PGE_2_ and COX-2 expression. EEAKSs (10 µg/mL) were treated in the media with DPB-LPS or PG-LPS (1 µg/mL) and the PGE_2_ and COX-2 level was estimated using the ELISA kit. ** *p* < 0.001 vs. only DPB-LPS treated cells, ^#^
*p* < 0.05, * *p* < 0.01 vs. only PG-LPS treated cells. (+), add; (-), no add. (**C**) The effect of EEAKSs on human cell growth. Immortalized human oral keratinocytes (IHOKs), immortalized human gingival fibroblasts (IGFs), and RAW264.7 macrophage cells were treated with indicated concentration of EEAKSs for 24 h and MTT assay was performed. DMSO was treated as the assay control.

**Figure 3 nutrients-14-00136-f003:**
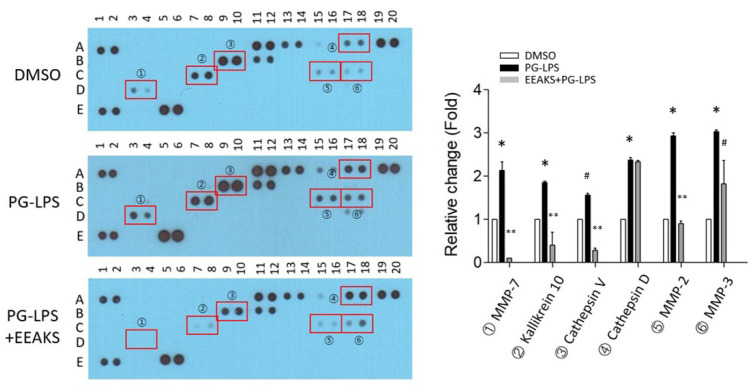
Effects of EEAKSs on PG-LPS-induced proteases levels. PG-LPS (1 µg/mL) treated in IHOK cell culture medium with or without EEAKSs (10 µg/mL) and culture medium (CM) was applied onto protease antibody array membrane. DMSO was treated as the control. Altered factors are indicated in the images as red rectangles and circled numbers. The list altered factor is shown as fold-change in the graph. * *p* < 0.01 vs. DMSO-treated medium, ** *p* < 0.001 vs. PG-LPS-treated medium, ^#^
*p* < 0.05 vs. PG-LPS-treated medium.

**Figure 4 nutrients-14-00136-f004:**
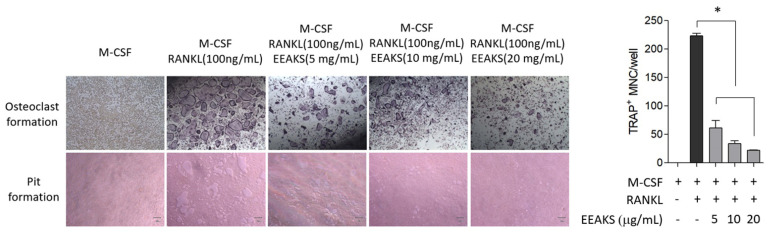
Effect of EEAKSs on RANKL-induced osteoclastic bone resorption. Mouse BMM cells were treated with M-CSF (30 ng/mL) and RANKL (100 ng/mL) with or without EEAKSs (5 or 10 µg/mL) and were stained to detect the expression of TRAP. Resorptive pit was also detected on calcium phosphate apatite-coated plates under light microscopy. The total number of TRAP-positive multinucleated (≥3 nuclei) osteoclasts (MNCs) per well is graphically presented. The data are expressed as the mean ± standard error of the mean (SEM). * *p* < 0.001 vs. M-CSF+RANKL condition. (+), add; (-), no add.

**Figure 5 nutrients-14-00136-f005:**
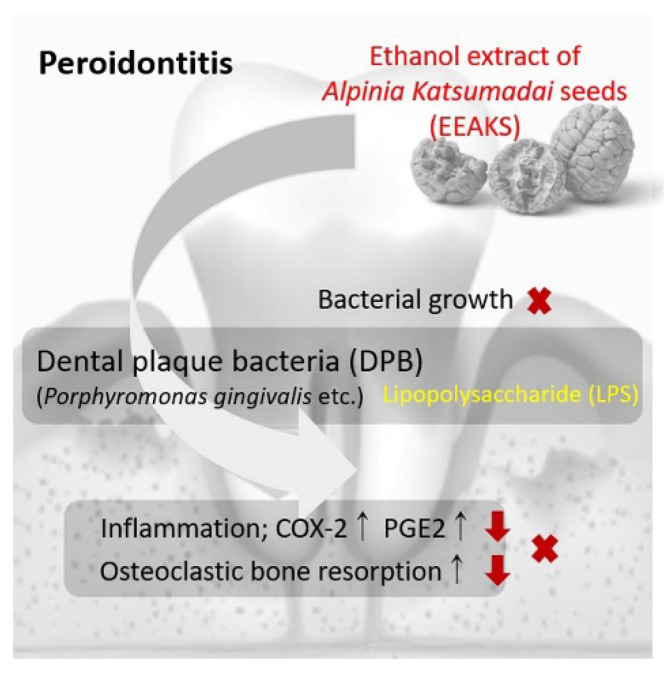
The proposed mechanism of ethanol extract of *Alpinia Katsumadai* seeds’ (EEAKSs’) anti-periodontitis effects.

## Data Availability

The data and materials of this article are included within the article.

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
