# Peer review of "Anti-Periodontitis Effect of Ethanol Extracts of Alpinia Katsumadai Seeds"

_nutrients, 2021, doi:10.3390/nu14010136_

Round 1
Reviewer 1 Report
Four individual effects of Alpinia Katsumadai-seed extracts (AKS) were demonstrated. (1) AKS directly inhibited bacterial growth of one P. gingivalis reference strain in a dose-dependent manner. (2) AKS inhibited secretion of inflammatory factors PGE2, COX-2 from gingival fibroblast-, oral keratinocyte-, and macrophage-cell lines, induced by LPS from a mixture of bacteria grown aerobically, or from P. gingivalis grown anaerobically. (3) AKS inhibited secretion of a number of proteases from oral keratinocytes induced by LPS of P. gingivalis. (4) AKS inhibited osteoclast differentiation induced by RANKL in macrophages and pit formation.
The manuscript is neatly written, although several missing descriptions need to be presented.
- Introduction, line 49: Please cite the reference where evidence of the statement in lines 49 to 50 “inflammatory mediators stimulated the secretion of collagenase from bacteria” is shown.
- The method for the ethanol extraction of Alpinia Katsumadai seeds (EEAKS) is missing. The information on the origin from where the AKS were purchased is also missing.
- Are the physiological effects of the EEAKS in the present study equivalent to the commercially available Alpinia Katsumadai-seed preparations? Please show if there are any comparisons tested for the anti-inflammatory or bacteriostatic effects.
- Materials and methods, 2.2. Bacterial culture and preparation: Please explain the reason why only aerobic cultivation was chosen to grow the dental plaque bacteria (DPB) in lines 85-86.
- Ethical agreement of the participant whom the DPB were obtained is missing, in line 88.
- Materials and Methods, 2.4. Cell lines and culture media: Please give the product names or numbers of the gingival fibroblasts (IGF) and human oral keratinocytes (IHOK), if available.
- Results, 3.1. Effects of EEAKS on P. gingivalis growth: One P. gingivalis reference strain was shown to be susceptible in growth inhibition by EEAKS. Is the growth inhibition effect of EEAKS species or genus specific?
- Results, 3.2. Effects of EEAKS on bacterial LPS-induced PGE2 and COX-2 levels: What is the difference between the origin of DPB-LPS-1, -2, -3?
Author Response
Comments and Suggestions for Authors
The manuscript is neatly written, although several missing descriptions need to be presented.
- Introduction, line 49: Please cite the reference where evidence of the statement in lines 49 to 50 “inflammatory mediators stimulated the secretion of collagenase from bacteria” is shown.
; Reference is cited (#4).
- The method for the ethanol extraction of Alpinia Katsumadai seeds (EEAKS) is missing. The information on the origin from where the AKS were purchased is also missing.
; Extraction method and source information are provided in 2.3 of Materials and Methods.
- Are the physiological effects of the EEAKS in the present study equivalent to the commercially available Alpinia Katsumadai-seed preparations? Please show if there are any comparisons tested for the anti-inflammatory or bacteriostatic effects.
; We did not use commercially available powder from Alpinia Katsumadai seeds in our experiments.
- Materials and methods, 2.2. Bacterial culture and preparation: Please explain the reason why only aerobic cultivation was chosen to grow the dental plaque bacteria (DPB) in lines 85-86.
; We also think that it would have been good to confirm the antibacterial activity of EEAKS by culturing dental plaque bacteria (DPB) under aerobic and anaerobic conditions. We think this is a limitation of our study. However, Pophyromonas gingivalis, an aggressive periodontitis-causing bacterium, has been confirmed to have antibacterial activity through anaerobic culture, thus partially proving that EEAKS inhibits the growth of anaerobes. The limitations of the study were described in the end of Discussion part.
- Ethical agreement of the participant whom the DPB were obtained is missing, in line 88.
; This experiment was conducted with ethical consent from the participants, and all experimental procedures were also approved by the IRB. Approval is specified in the ‘Ethics approval and consent to participate’ section in almost end of manuscript.
- Materials and Methods, 2.4. Cell lines and culture media: Please give the product names or numbers of the gingival fibroblasts (IGF) and human oral keratinocytes (IHOK), if available.
; These were established directly in the Oral Cancer Institute in Yonsei University College of Dentistry that provided the cells. We can provide the cells if the use of cells is appropriate. See 2.1 Cell lines and culture media in Materials and Methods.
- Results, 3.1. Effects of EEAKS on gingivalisgrowth: One P. gingivalis reference strain was shown to be susceptible in growth inhibition by EEAKS. Is the growth inhibition effect of EEAKS species or genus specific?
; We obtained one P. gingivalis strain isolated from the human dental plaque in Cell Bank and used it in the experiment. Therefore, we did not analyze according to species or genus.
- Results, 3.2. Effects of EEAKS on bacterial LPS-induced PGE2 and COX-2 levels: What is the difference between the origin of DPB-LPS-1, -2, -3?
; We cultured bacteria from the dental plaques of 3 participants, extracted LPS, and labeled them as DPB-LPS-1, DPB-LPS-2, DPB-LPS-3.
We are pleased to have the opportunity to revise our manuscript.
Referring to your duplication report for our manuscript, we corrected the part with high similarity. In particular, the Material and Methods section has a high similarity to the manuscript we published in 2017 (DOI 10.1186/s12906-017-1619-1), so we focused on the revision.
The corrected part is indicated in red letters.
Reviewer 2 Report
In my opinion the manuscript entitled „Anti-periodontitis effect of ethanol extracts of Alpinia Katsumadai seeds” presents an interesting data, but manuscript requires major corrections. This study was carried out correctly, also material and methods are appropriate.
I have a few comments:
- Abstract – needs improvement, the names of the chemicals should be removed.
- The Introduction is too short - please expand this section. Moreover, Authors should provide more information on the current state of knowledge. What exactly is known about the extract of Alpinia Katsumadai seeds? Why was this extract chosen?
- Please provide a study scheme.
- Figures should be placed in the Results section.
- Discussion needs major improvements. In the current version it is more like the introduction. Authors repeat the information from the introduction to the manuscript.
- Please also indicate the limitations of the study.
Author Response
I have a few comments:
- Abstract – needs improvement, the names of the chemicals should be removed.
; Proved as appropriate as possible.
- The Introduction is too short - please expand this section. Moreover, Authors should provide more information on the current state of knowledge. What exactly is known about the extract of Alpinia Katsumadai seeds? Why was this extract chosen?
; Proved as appropriate as possible. The reason for choosing Alpinia Katsumadai seeds extract is described in the second paragraph of the Discussion (in red, line 224-229).
- Please provide a study scheme.
; It is provided in Figure 5.
- Figures should be placed in the Results section.
; We wrote the manuscript according to the journal’s manuscript guidelines.
- Discussion needs major improvements. In the current version it is more like the introduction. Authors repeat the information from the introduction to the manuscript.
; Proved as appropriate as possible.
- Please also indicate the limitations of the study.
; It is a limitation of our study that the antibacterial and anti-inflammatory activities of EEAKS were not observed by culturing dental plaque bacteria aerobically and anaerobically. In addition, if analytical grade EEAKS powder are commercially available, they can be used as an experimental control for analysis. This content is described in the end of Discussion part.
We are pleased to have the opportunity to revise our manuscript.
Referring to your duplication report for our manuscript, we corrected the part with high similarity. In particular, the Material and Methods section has a high similarity to the manuscript we published in 2017 (DOI 10.1186/s12906-017-1619-1), so we focused on the revision.
The corrected part is indicated in red letters.
Round 2
Reviewer 2 Report
In my opinion, the figures should not be at the end of the manuscript.
Author Response
We have edited the manuscript appropriately according to the comments.
"Please see the attachment."
